# Application of Cancer Organoid Model for Drug Screening and Personalized Therapy

**DOI:** 10.3390/cells8050470

**Published:** 2019-05-17

**Authors:** Jumpei Kondo, Masahiro Inoue

**Affiliations:** Department of Clinical Bio-resource Research and Development, Graduate School of Medicine Kyoto University, Kyoto 606-8501, Japan; masa_inoue@kuhp.kyoto-u.ac.jp

**Keywords:** drug screening, cancer, cell lines, organoid, spheroid, heterogeneity, precision medicine

## Abstract

Drug screening—i.e., testing the effects of a number of drugs in multiple cell lines—is used for drug discovery and development, and can also be performed to evaluate the heterogeneity of a disease entity. Notably, intertumoral heterogeneity is a large hurdle to overcome for establishing standard cancer treatment, necessitating disease models better than conventional established 2D cell lines for screening novel treatment candidates. In the present review, we outline recent progress regarding experimental cancer models having more physiological and clinical relevance for drug screening, which are important for the successful evaluation of cellular response to drugs. The review is particularly focused on drug screening using the cancer organoid model, which is emerging as a better physiological disease model than conventional established 2D cell lines. We also review the use of cancer organoids to examine intertumor and intratumor heterogeneity, and introduce the perspective of the clinical use of cancer organoids to enable precision medicine.

## 1. Introduction

Drug screening—using variable assays to test multiple candidate drugs or compound libraries—is a method to obtain hit drugs that may potentially achieve the desired objectives. However, when performing drug screening for drug discovery and development, enormous amounts of money and time must be spent to obtain clinically approved drugs [1,2,3,4,5,6]. To obtain a single approved drug, tens of thousands of compounds are generally put through several screening stages prior to clinical trials. Even after the long and costly process to identify lead compounds (drug discovery) and generate optimized derivatives (lead optimization), ~80% of drugs fail during clinical trials. Wong et al. analyzed 406,038 clinical trial data entries for over 21,143 compounds from January 1, 2000 to October 31, 2015 [7], and found that the overall success rate of phase I–III clinical trials was 13.8%, with an extremely low success rate for cancer treatment (3.4%) and a 20.9% success rate for all the other entries.

Why do so many clinical trials fail? A series of studies analyzed failures in phase II and phase III clinical trials for the time periods of 2007–2010 [5] and 2013–2015 [6], and reported that the most common reason for failure was lack of efficacy (56% and 52%, for each period respectively), followed by safety issues (28% and 24%, respectively). In addition to biological reasons, studies also failed due to inadequate study design, including the selection of the dose, efficacy markers, and schedule, as well as data analysis problems. However, such causes were less common, with 7% (2007–2010) and 15% (2013–2015) of failures related to strategic factors, and 5% (2007–2010) and 3% (2013–2015) related to operational factors. These findings highlight the importance of developing robust systems to predict actual clinical efficacy during the drug screening steps. In particular, since cancer is a highly heterogeneous disease, accurate prediction of efficacy is critical to achieve novel approved treatments.

In this review, we outline the recent progress in using experimental cancer models to screen for drugs with greater physiological and clinical relevance. We particularly focus on details of the cancer organoid model, which is emerging as a better physiological disease model than conventional established 2D cell lines.

## 2. Screening System for Cancer Drug Discovery

A drug screening system comprises three main components: compounds or drugs to be screened, the screening methods, and the materials to be screened. Different factors can be combined to develop an appropriate screening system to best meet the aim of the screening project. Advances in each component contribute to the overall improvement of screening systems. In recent years, drug repositioning—the concept of re-developing previously approved or discontinued drugs for novel indications—has attracted attention as a means of saving cost and time in new drug development [8,9,10,11]. Additionally, there is growing interest in screening aimed at identifying combination therapy that may overcome resistance to targeted therapies. Advances in high-throughput screening systems have allowed the evaluation of tens or hundreds of thousands of compounds/drugs, and the narrowing down of potential candidates, with the use of automated machines to dispense cells and drugs, and to execute endpoint assays [12,13]. In silico methods have also become important in drug discovery and drug repositioning [14,15].

In addition to advances in compounds/drugs and screening methods, cancer models as materials to be screened have remarkably improved over the past decade (Table 1). Historically, the only materials for cancer drug screening have been cultured established cancer cell lines in two-dimensional (2D) culture. Such established cell lines are often readily obtainable from cell banks, such as the American Type Culture Collection (ATCC), and can be maintained using standardized culture method. In contrast, biomaterials are more difficult to obtain, and their handling is too complex to be suitable for high-throughput screening.

The methods used for 2D culture of established cell lines enable their preparation in large quantities at uniform quality, which is necessary for high-throughput drug screening. However, 2D cultured cells exhibit a flattened morphology and altered signaling networks compared to cancer cells in tumor tissue in vivo, which unwantedly affects their response to drugs [16]. Additionally, when cells form 3D cell clusters or spheroids, they sometimes acquire resistance to anti-cancer drugs, known as multi-cellular resistance [17]. Culture medium containing serum for conventional 2D culture of cancer cells induces cell transformation and accelerates clonal selection, resulting in loss of the parental tumor’s characteristics [18,19,20]. Thus, disease models or materials that better reflect the physiological features of an actual patient’s cancer, have long been sought for cancer research. In the next section, we review recent advances in cancer disease models.

## 3. Developments and Advances in Disease Models for Cancer Research and Drug Screening

To better recapitulate patients’ cancer, various disease models have been developed. We feature three models in this chapter; 3D culture of established cell lines, animal models, and organoid models to discuss the current arguments of 2D vs. 3D, in vivo vs. ex vivo, and established cell lines vs primary or early passaged cells. Due to space limitations, we do not touch other models which might also have preclinical potentials; conditionally reprogrammed cell lines [21,22], early passage PDX 2D cell lines [23], co-culture with fibroblast [24] and immune cells [25], and explant/organotypic culture models [26,27].

### 3.1. Three Dimentional (3D) Culture of Established Cell Lines

Cell characteristics are largely affected by the architecture of the cell population, e.g., cell–cell contact, cell–matrix contact, and the polarity. Oxygen and nutrients are diffused along a concentration gradient within the 3D architecture. Thus, it is becoming widely accepted that 3D cell cultures better reflect the physiological features of the cells compared to 2D cultures (Table 1) [28,29,30,31]. The simplest 3D architecture is a spherical aggregate of cells, called a spheroid. Some conventional 2D cell lines can be induced to form spheroids by simply seeding them in a low-attachment round-bottom plate. However, since most 2D established cell lines are optimized for culture as adherent cells, they tend to adhere to the bottom of an untreated plastic surface.

Newly developed devices and techniques can promote conventional cell lines to form spheroids or other types of 3D architecture. Such techniques include relatively traditional methods, such as cell culture in a spinner flask, rotation culture, and the hanging drop method [32,33]. Moreover, biological or synthetic scaffolds that provide anchorage for the cells can be used for efficient 3D culture of established cell lines [33]. Relatively easy systems for 3D cell culture involve collagen or laminin matrices [34]. Another more complex approach includes the re-establishment of microenvironments by providing synthesized matrices together with co-cultured stromal and/or endothelial cells [32,33,35]. Despite wide use of 3D cultured traditional cell lines, researchers must consider that the original tumor characteristics lost in the established cell lines are not entirely recovered by the forced 3D architecture [30].

### 3.2. Animal Models

There are two main types of animal models: genetically modified models (GEM) and patient-derived xenografts (PDX), in which a piece of a patient’s tumor is implanted in immune-deficient mice (Table 1). Both models are used to study disease pathology and biology, as well as for pre-clinical validation of candidate drug efficacy and safety [36,37,38]. Compared to others, animal models have the robust advantage of enabling evaluation of the candidate drugs’ efficacy within a relatively physiological environment comprising both the stroma and the host cells, including fibroblasts, endothelial cells, blood cells, and immune cells. Such models enable the evaluation of pharmacodynamics and pharmacokinetics.

GEM provide a good platform for analyzing the molecular mechanisms of cancer, and obtaining proof of concept for molecular targeting drugs. However, GEM is less useful for reconstituting and evaluating the heterogeneity of actual patient populations. Compared to established 2D cell lines, the cancer cells in PDX show greater preservation of the original tumor characteristics [38,39,40,41,42]. Moreover, even after serial passages, PDX are thought to exhibit only minimal genetic drift from the original tumor [39,43,44,45,46], although it has also been reported that genetic changes, such as copy number alteration, can accumulate over multiple passages [47].

Recent success in immune checkpoint therapy—for example, using PD-1/PD-L1 and CTLA-4 inhibitors [48,49]—has increased the importance of using model systems to evaluate immune responses in vivo. PDX are generated in immunodeficient mice, making immune response evaluation impossible. GEM with murine immune systems also have limitations, as they do not accurately recapitulate the human immune system. These issues have led to the development of humanized mice, generated by transplanting human hematopoietic stem cells or immune cells into immunodeficient mice, to reconstitute the human immunologic environment [50,51]. Several reports describe the usefulness of PDX models in humanized mice for evaluating the efficacy of immunotherapy [52,53,54]. Although animal models are more physiological than in vitro models, using animal models for high-throughput drug screening is highly expensive and time-consuming. Some reports demonstrate the usefulness of drug testing in either mid-size (~100 animals) [55] or large (~1,000 animals) [41] cohorts of PDX models, at relatively low throughput (up to ~20 single or combined regimens) [41,55,56,57]. Such protocols might be compatible with phase I–II trial studies. Overall, GEM and PDX are robust preclinical models with physiological microenvironment, which can be hardly reproduced by ex vivo models. On the other hand, the time and cost required make them difficult to apply in early stages of drug development, which require the screening of large numbers of candidate drugs.

### 3.3. Organoid Models

An organoid is defined as “a 3D structure developed or grown from stem cells and consisting of organ-specific cell types that self-organizes through cell sorting and spatially restricted lineage commitment in a manner similar to in vivo.” [58,59]. This definition has been derived from the fields of developmental and stem-cell research [58,59,60,61], and is extended to cultured cancer cells using similar methods (Table 2).

#### 3.3.1. Cancer Stem-Like Cell Organoid

Current cancer organoid culture methods can be traced back to neurosphere cultures reported in 1992 by Reynolds and Weiss, in which neural stem cells in serum-free defined medium were able to proliferate as floating spheroids that retained the potential to develop into neurons and astrocytes [62]. The medium used for neurosphere culture was DMEM/F12 supplemented with EGF, insulin, transferrin, progesterone, putrescine, and selenium salt. Subsequently, this method was applied to the culture of normal mammary gland and breast cancer cells as 3D mammospheres and organoids, respectively, which better reflect the physiological characteristics of these cells in their original environment compared to conventional 2D culture with serum [20,63,64].

In the field of cancer research, tumor spheroid culture was used in early reports of cancer stem-like cells—a tumorigenic population of primary cancer cells that express distinct cancer-stem markers. Cancer stem-like cells derived from patients’ tumors exhibited efficient spheroid formation and could be maintained for a long time in serum-free stem cell medium (a slight modification of Reynolds’ neurosphere medium) [65,66,67]. Notably, when cultured in this serum-free stem cell medium, cancer cells could be maintained without the restriction of Hayflick limit, which previously thought to limit the long-term culture of primary cancer cells.

More recently, the 3D culture of primary cells was robustly advanced by the invention of a method to culture normal epithelial stem cells as organoids. WNT pathway activation by R-spondin1, together with the inhibition of BMP pathway (the activation of which induces stem cell differentiation), enabled the cultivation of small intestinal stem cells as 3D organoids [68]. This technical breakthrough was subsequently applied to the culture of stem cells from many other types of normal epithelial tissues, including colon [69], liver [70,71], pancreas [72], prostate [73], uterus [74,75], and more [59], as well as the cancer cells of these tissues [59].

Based on these advances in culture systems, the 3D culture of primary cancer cells as spheroids or organoids is now a widely accepted practical disease model. Compared to 2D or 3D cultured cell lines, primary cancer organoids have the advantage of retaining the characteristics of the cancer cells from the original patients’ tissue, including their differentiated morphology and responses to drugs [18,19,20,59]. Moreover, primary cancer organoid cells accumulate much fewer genetic alterations over multiple passages compared to cells in 2D culture with serum-containing medium. Organoid culture can also be readily combined with biomaterials, and applied to an ex vivo platform for a PDX system.

The organoid model also currently has several drawbacks. Organoids are difficult to generate due to multiple technical and logistical factors. Technically, their generation still requires craftmanship to prepare primary cells from individual patients’ tissue. Logistically, research institutes without affiliated hospitals will have poor access to the necessary materials. Although several biobank resources are being established to distribute patient-derived cancer organoids [76,77,78,79], they are not yet as abundant or easily accessible as 2D cell lines.

#### 3.3.2. Cancer Tissue-Originated Spheroid (CTOS) Method

Briefly, cancer organoids are prepared by dissociating tumor tissue into single cells, collecting these cancer cells, and letting them form spheroids either in suspension or in matrices (Figure 1). However, unlike established cell lines, single cells prepared from primary tumor tissues are prone to undergoing anoikis [30], which is a distinct form of apoptosis triggered by the loss of anchorage [80]. Anoikis rapidly begins upon dissociation of tissue or cell clusters, thus the dispersed cells must immediately adhere to the plastic surface of the culture dish or become embedded in matrices. Rho-ROCK inhibitor suppresses anoikis [81,82], and improves the efficiency of primary organoid and spheroid preparation [83,84,85].

Our group has also developed the cancer tissue-originated spheroid (CTOS) method, which is a distinct technique for preparing and culturing organoids without ever dispersing the cancer cells into single cells (Table 1) [30]. In the CTOS method, cell–cell interactions are maintained between cancer cells during the enzymatic digestion of cell–matrix interaction, and the subsequent culturing (Figure 1). A tumor specimen is digested, and the resultant tissue fragments and cells are passed through mesh filters. The fragments trapped by the filters are used for further culture. The epithelial cancer tissue fragments rapidly form spheroids (CTOSs) with minimal cell death because anoikis is not triggered [30]. This process enables a high recovery rate of the primary cancer cells from tissue. Moreover, the CTOS method enables easy collection of the population of purified cancer cells, since the blood cells, fibroblasts, and dead cancer cells pass thorough the filter. Overall, the CTOS method allows the preparation of organoids from a primary tumor with high viability, efficiency, and purity. We and others have reported the establishment and use of organoid cultures via the CTOS method in multiple types of cancers, including colorectal [30,86,87,88,89,90], lung [91], urothelial [92,93,94,95,96], liver [97], head and neck [98], and uterine [99,100,101] cancers. The drawbacks described for CSC-organoids are also applicable to CTOS organoids; technical difficulties and limited access to patient samples are the issues to be overcome.

## 4. Drug Testing and Screening for Cancer Drug Discovery and Personalized Medicine using Cancer Organoids

### 4.1. Cancer Stem-Like Cell Organoids

Patient-derived 3D cancer organoids are increasingly being studied for potential use as an improved platform for drug discovery and personalized medicine. The number of reports on this topic have particularly increased since 2017 (Table 2). Clevers’ group reported the use of 19 colorectal cancer (CRC) organoid lines to screen 83 drugs, including target-known inhibitors and chemotherapy drugs [76], and the use of 28 breast cancer organoid lines to screen 6 EGFR/AKT/mTORC pathway inhibitors [78]. The same methodology was also applied to screen 37 drugs using organoids from 7 gastric cancer patients [77], and 26 drugs using 11 bladder cancer organoid lines [102]. Importantly, these organoid lines exhibited interpatient and intratumor heterogeneity in their drug responses, representing proof of the concept that cancer organoids are a useful platform for drug screening.

Li et al. reported the screening of 129 cancer drugs using 27 liver cancer organoid lines derived from 5 different patients [103]. They found that some drugs were highly heterogeneous in the responses among organoid lines, while others were uniformly toxic in all cases. Pauli et al. performed relatively higher-throughput screening of 160 drugs using cancer organoids from 4 patients, and combination drug screening for hit drugs plus 120 candidates [104]. They screened organoids in 2D culture, probably to circumvent the cumbersome process for culturing 3D organoids. They reported that only ~5% of the tested drugs showed preferential activity when assayed in 2D versus 3D. On the other hand, Jabs et al. screened 22 single or combination treatments using 10 ovarian cancer organoids, and reported that drug responses were more diverse in cancer organoids than in 2D cultured cells [105]. Moreover, patient-specific genomic alterations correlated with drug effects observed in organoids but not in 2D cell monolayers, supporting that 3D organoids are a better physiological model than 2D primary cells. Further accumulation of experiments comparing 3D with 2D models is necessary for revealing superiority in actual translational value or predictive power.

These preliminary reports encourage the use of cancer organoids as a platform to study heterogeneity in the cancer population. The link between ex vivo sensitivity and patient outcome remains to be further validated.

### 4.2. CTOS Organoids

The above-described examples of drug screening using cancer organoids are relatively low throughput, with greater focus on the aspect of personalized medicine than on early stage of drug discovery. To conduct drug screening, especially high-throughput screening, the amount of available cells is a critical factor. To overcome this issue, the CTOS method can include the generation of xenograft tumors, and the preparation of organoids from these xenografts. The high recovery rate and highly purified cancer cells produced by the CTOS method enable the generation of organoids ready for drug screening briefly after preparation [42].

Following this strategy, we have reported the use of CTOS organoids for drug screening (Table 2) [42,87,100]. We employed the ‘2-step scheme’; first the screening of a drug library is performed on few CTOS organoids from different patients, and obtain hit drugs, and then the effect of the hit drugs are evaluated in a larger number of CTOS lines to assess the diversity of the response to the hit drugs. As a proof of concept study, we first screened a drug library of ~100 inhibitors using CTOS organoids from 2 cases of endometrial cancer [100]. The putative survivin inhibitor sepantronium bromide, which is widely known as YM155, was identified as a hit drug that inhibited the growth of both cervical CTOS organoids. Subsequent analysis of YM155 using cervical CTOS organoids from 12 different cases revealed that the sensitivity largely varied among cases, and that the sensitivity to YM155 was correlated with cancer histopathology. We conducted this small-scale drug screening in cervical cancer using manual handling of CTOS organoids with a micropipette under a microscope, and manual dispensing of the tested drugs.

To evaluate thousands of drugs by high-throughput screening, automated systems are needed to dispense organoids and reagents. Thus, we used automated machines—an organoid handler and a reagent dispenser—to screen a library of 2427 drugs (including FDA-approved drugs and small molecules with known targets) with CTOS organoids from 2 cases of CRC [42]. Among these 2427 drugs, 15 (0.6%) hit drugs suppressed the viability of either CRC CTOS line, of which 8 drugs were effective against both lines. Ten of these hit drugs were anti-cancer reagents, and 7 were already approved by the FDA. We then evaluated sensitivity to the 15 hit drugs in a panel of CRC CTOS lines from 30 different cases, and found that these different lines showed diverse sensitivities to the hit compounds. Notably, the variation in drug sensitivity among 30 CRC CTOS lines was much broader than that among 39 established cell lines of different cancer types, demonstrating the usefulness of CTOS organoids with the 2-step scheme for investigating the highly heterogeneous nature of cancer. These studies demonstrate that the 3D cancer organoid model prepared and cultured with the CTOS method is a robust tool for both high-throughput drug screening and pre-clinical testing.

### 4.3. Validation of the In Vitro Assay System

Validating the clinical relevancy of the in vitro assay system is critically important; however, there are multiple issues related to collecting live cells from patients’ tumors for this purpose. The main source of tissue for organoid generation is excess surgical specimen. Patients typically do not receive intensive drug therapy after complete tumor resection. Even patients who receive adjuvant therapy usually do not have lesions to be evaluated for the drug response, such that progression-free survival or overall survival are the only possible end-points. Moreover, the tumors at recurrence are not necessarily the same tumors from the initial surgical resection. Notably, unnecessary biopsy must be avoided for patients scheduled for standard therapy. Thus, the physiological relevancy is only validated by the comparison between in vitro and in vivo assays.

Schütte et al. established a CRC biobank comprising organoids and xenografts derived from a cohort of 106 patients [106]. They compared the treatment outcomes with 8 drugs in 19 pairs of organoid/PDX siblings. Among the 152 combinations of treatments and cases, 18 combinations (11.8%) showed discordant responses. The gene expression profiles of the EGFR blockade response shared only a few genes between PDX and organoids, highlighting their biological differences and the difficulty of directly comparing these systems.

Practical optimizations are needed to ensure a close correlation between the sensitivity assay result and the actual clinical outcome, as well as to overcome the presently limited access to clinical specimens before non-surgical treatment. Moreover, optimization of the sensitivity assay should also be applied to the drug screening method, to reduce false-negative or false-positive selection of candidate drugs. Thus, it is desirable to develop an improved method that will be applicable for both drug screening and personalized medicine [7,9,107].

### 4.4. Perspective for the Use of 3D Organoid Culture in Personalized Medicine

As described above, the majority of hit drugs show inter-tumor heterogeneity, while others are uniformly toxic in all cases. Thus, it is critically important to evaluate the inter-tumor heterogeneity to estimate the responders to a hit drug in clinical use. Despite obvious interpatient heterogeneity regarding sensitivity to anti-cancer drugs in clinical use, this perspective has been neglected in most previous reports, especially in studies using 2D established cell lines. Prior drug development might have been conducted under the implicit assumption that cancers of the same pathological type uniformly respond to a drug, or drug developers may have been seeking a “magic bullet” to overcome the heterogeneity. Regardless, except for some molecular-targeted drugs that are designed to attack certain pathway mutations, clinical studies of most drugs under development are conducted without any biomarkers, or even data regarding the predicted ratio of potential responders. Therefore, it is critical to develop methods to predict drug sensitivity, and thus reduce failure in clinical trials. One useful strategy may be to employ panels of cancer organoids, which are expected to reflect the intertumor heterogeneity, to evaluate variations in drug responses and to discover companion biomarkers for the drugs under development. One sophisticated way to predict drug sensitivity is based on mutational information, such as EGFR mutation for the use of EGFR tyrosine kinase inhibitors in lung adenocarcinoma [108,109]. Theoretically, advances in gene sequencing technologies enable screening for genetic predictors with potential clinical utility, even during the clinical development of a new drug. Indeed, trials using biomarkers are almost twice as successful compared to trials without biomarkers (10.3% vs. 5.5%) [7]. However, to date, only a small proportion of drugs have known genetic predictors with clinically meaningful effects [110]. The concept of functional precision medicine, or functional biomarkers, is attracting attention as an alternative way to predict drug efficacy [111]. Rather than exclusively relying on a snapshot of information, functional precision medicine will offer highly actionable and functional information from assessment of the characteristics of viable primary tumor cells. Genetic analyses and drug sensitivity testing using cancer organoids could be complementary methods for predicting effective drugs before treatment.

Intratumor heterogeneity can be another concern for the personalized medicine using patient derived live materials. This is a general issue of sampling from the tumor, because populations of tumor cells in different part of the same tumor might have different drug sensitivity. Practically, are multiple samplings necessary? How many samplings are enough? These questions will be answered when personalized medicine based on the tests using patient derived cells is realized in clinic.

## 5. Conclusions

The use of cancer organoids, as both models and methods, is expanding in the fields of basic and preclinical cancer research. Drug screening systems using cancer organoids could potentially be applied to determine the most beneficial drug for each individual patient. Over the next decade, research should address the optimization of the organoid culture method to improve its efficacy and feasibility, and improve the concordance between drug responses and actual clinical outcome. Moreover, there is a need to establish biobanks that can provide organoids as resources to support their wide use as a standard platform of cancer research.

## Figures and Tables

**Figure 1 cells-08-00470-f001:**
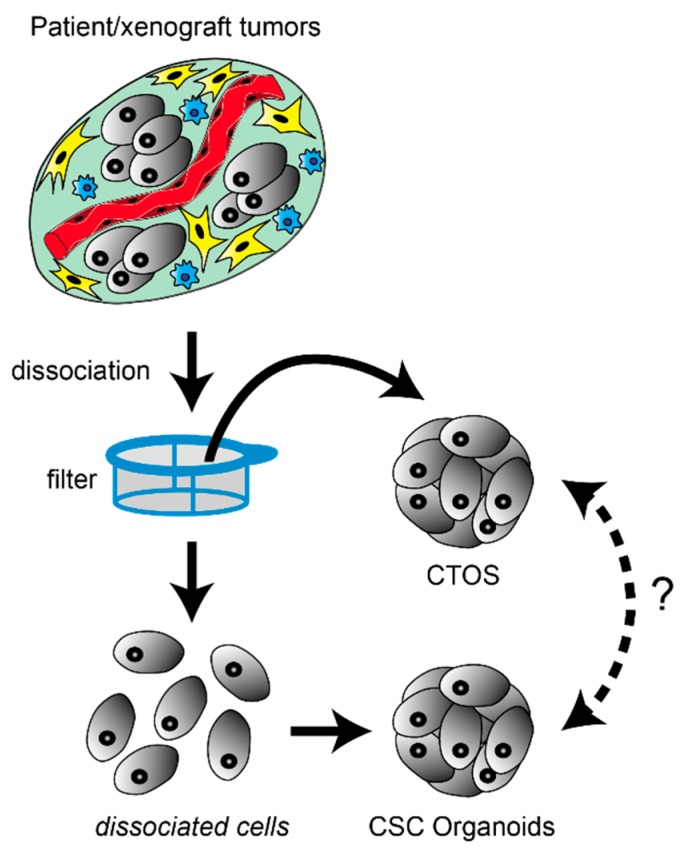
Preparation of cancer stem-like cell (CSC)-derived organoids and cancer tissue-originated spheroids (CTOSs). In both methods, tumor specimens are mechanically and enzymatically digested. To generate CSC-derived organoids, tumor tissues are dispersed at the single-cell level, and organoid formation occurs by either re-aggregation or clonal proliferation. Optionally, incompletely dissociated fragments trapped by filters are further trypsinized to recover more single cells. In contrast, to generate CTOSs, tumor tissues are incompletely digested into fragments on purpose by milder conditions. These cell clusters rapidly form spherical-shaped CTOSs within 24 h. Because the cell-cell contact is retained during the preparation process, CTOS could better recover heterogenous cells including those at non-CSC state by avoiding anoikis, On the other hand, CSC-derived organoids could better enrich the cells with cancer-stem like property. However, it remains an open question whether the cells obtained by these different methods are interchangeable or not.

**Table 1 cells-08-00470-t001:** Pros and cons of disease models.

Model		Accessibility	Feasibility	Intertumor Heterogeneity	Intratumor Heterogeneity	Physiological Characteristics	Applicability to HTS
Cell lines	**2D**	Good	Good	Allows comparison between cell lines	Poor	Largely lost	Good
	**3D**	Good	Complex in some systems with biomaterials	Allows comparison between cell lines	Poor	Partially reestablished	Difficult for some cell lines
Animal models	**GEM**	Relatively good once generated	Laborious for double or triple GEMs	Partially allows comparison	Good	Good, including microenvironment and immune system	Not suitable for HTS
	**PDX**	Requires access to hospital or tissue network	Good once established	Allows comparison between multiple cases	Good	Good, including microenvironment	Not suitable for HTS
Organoids	**CSC-derived organoids**	Requires access to hospital or tissue network	Requires skills, may suffer from low recovery rate	Allows comparison between multiple cases	Good (may select for cells resistant to anoikis)	Good	Possible but costly
	**CTOS organoids**	Requires access to hospital or tissue network	Requires skills	Allows comparison between multiple cases	Good	Good	Good as an ex vivo setting

HTS, high-throughput screening; GEM, genetically modified models; PDX, patient-derived xenografts; CSC, cancer stem cell; CTOS, cancer tissue-originated spheroid.

**Table 2 cells-08-00470-t002:** Summary of reports of drug screening with organoid methods.

Cancer type	Organoid Type	Library	# Compounds Tested	# Cases Tested	Assay Conditions	Reference
Colorectal	CSC-derived	Target-known inhibitors + chemo drugs	83	19	With 2% BME in culture medium on BME	[76]
Breast	CSC-derived	EGFR/AKT/mTORC pathway inhibitors	6	28	With 2% BME in culture medium on BME	[78]
Gastric	CSC-derived	Approved anti-cancer drugs	37	7	On 50% Matrigel	[77]
Bladder	CSC-derived	Target-known inhibitors + chemo drugs	50	11	With 2% Matrigel in culture medium	[102]
Liver	CSC-derived	NCI-Approved Oncology Drugs Set VII	129	5	In Matrigel	[103]
Various	CSC-derived	Chemo drugs and targeted agents under clinical development	160 (single) + 120 (combination)	4	2D culture of organoids for screening	[104]
Ovarian	CSC-derived	Target-known inhibitors + chemo drugs	22	10	With 2% Matrigel in culture medium on Matrigel	[105]
Colorectal	CSC-derived *	Target-known inhibitors + chemo drugs	8	19	In Matrigel	[106]
Endometrial	CTOS	Target-known inhibitors	79	5 (2 hit drugs evaluated in 12 CTOS lines)	w/o matrix	[100]
Colorectal	CTOS	Target-known inhibitors	71	1	w/o matrix	[87]
Colorectal	CTOS	Target-known inhibitors + FDA-approved drugs	2427	2 (15 hit drugs evaluated in 30 CTOS lines)	w/o matrix	[42]

* Prepared by CTOS method. CSC, cancer stem cell; CTOS, cancer tissue-originated spheroid; BME, basement membrane extract.

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
