# Peer review of "Application of Cancer Organoid Model for Drug Screening and Personalized Therapy"

_cells, 2019, doi:10.3390/cells8050470_

Round 1
Reviewer 1 Report
In this manuscript, various types of patient-derived cancer models were comprehensively overviewed. Notably, the upsides of the CTOS method that the authors developed were illustrated in detail. However, there are a few points that need to be clarified before publication of this article.
Major points:
(1) In the sections 3 and 4, organoids and CTOS were separately documented, suggesting that they are distinct types of cellular population. In table2, however, in the column for "organoid type", CSC-derived and CTOS were included. This classification might point toward the notion that CTOS is one particular form of organoid, which contradict with the initial classification. The reviewer speculates that the term "organoids" might have been used in a strict sense or in a broad sense, depending on the context. However, it should be documented that way, or in a totally different way, to avoid any confusion with terminology. Given that the title of this article also contains "organoid", it should be mentioned somewhere in the manuscript that "organoid" could be used in both meanings.
(2) Figure1 provides schematic presentation of methodological differences between CTOS and organoids. However, it appears that critical information is lacking from this figure. By definition, CTOS derives from cell cluster that retained cell-cell contact, but not from single cells. Single cells were supposed be discarded according to the original protocol, because it did not form spheroids in floating culture. However, in this figure, it is shown that these singly dissociated cells could give rise to organoids. The only difference is in fact the presence of extracellular matrix such as Matrigel. With Matrigel, both cell clusters and single cells could develop organoids, while CTOS can be generated only from cell clusters without Matrigel. These explanations might facilitate the readers' understanding of each system.
(3) For high-throughput drug screening, the use of Matrigel may not be ideal. Consequently, researchers could turn organoids into floating spheroids for these types of assay. One interesting point is that, as the authors previously demonstrated, cellular polarity are opposite between organoids and spheroids. Is it known or not known whether it could affect drug sensitivity? Such information, if available, will be of technical value. Also, the fact that organoids and spheroids are interchangeable might as well be described in Figure1.
Minor points:
(1) Line 81 of Page2: "3.1.3. D culture" might be "3.1. 3D culture ".
(2) Line 69 of 5. Conclusion, "3.2. Figures and Tables" was erroneously inserted.
(3) All the references were numbered twice.
Author Response
We thank the reviewer for his/her constructive comments. We have carefully reviewed the comments, and respond as following. In this point-by-point response, to clearly distinguish our response from the reviewer’s comments, our responses were shown in blue texts. Also, in the main text, blue font was used to clearly indicate the revised portions of the manuscript.
Major Point 1: In the sections 3 and 4, organoids and CTOS were separately documented, suggesting that they are distinct types of cellular population. In table2, however, in the column for "organoid type", CSC-derived and CTOS were included. This classification might point toward the notion that CTOS is one particular form of organoid, which contradict with the initial classification. The reviewer speculates that the term "organoids" might have been used in a strict sense or in a broad sense, depending on the context. However, it should be documented that way, or in a totally different way, to avoid any confusion with terminology. Given that the title of this article also contains "organoid", it should be mentioned somewhere in the manuscript that "organoid" could be used in both meanings.
Response 1: We meant to describe CTOS as one particular form of organoid. To clearly indicate this, structure of chapter 3 was amended. Both the section 3.3.1) CSC-Organoid and 3.3.2) CTOS Method were described under the section 3.3) Organoid Model. Similarly, the titles of the sections in chapter 4 were amended to 4.1) CTC-Organoids and 4.2) CTOS Organoids.
Major Point 2: Figure1 provides schematic presentation of methodological differences between CTOS and organoids. However, it appears that critical information is lacking from this figure. By definition, CTOS derives from cell cluster that retained cell-cell contact, but not from single cells. Single cells were supposed be discarded according to the original protocol, because it did not form spheroids in floating culture. However, in this figure, it is shown that these singly dissociated cells could give rise to organoids. The only difference is in fact the presence of extracellular matrix such as Matrigel. With Matrigel, both cell clusters and single cells could develop organoids, while CTOS can be generated only from cell clusters without Matrigel. These explanations might facilitate the readers' understanding of each system.
Response 2: Indeed, cell clusters captured by filter and the single cell passed through the filters can be different. By avoiding anoikis, CTOS could better recover heterogenous cells including those at non-CSC state. On the other hand, CSC-derived organoids could better enrich the cells with cancer-stem like property. Actually, in our preliminary experiment, expression of surface markers such as CD133 differed between cell clusters trapped on filter and flow-through single cells. However, it remains an open question whether the cells obtained by these different methods are interchangeable or not. These insights are now described in the legend of Figure 1 to facilitate the readers’ understanding.
Major Point 3: For high-throughput drug screening, the use of Matrigel may not be ideal. Consequently, researchers could turn organoids into floating spheroids for these types of assay. One interesting point is that, as the authors previously demonstrated, cellular polarity are opposite between organoids and spheroids. Is it known or not known whether it could affect drug sensitivity? Such information, if available, will be of technical value. Also, the fact that organoids and spheroids are interchangeable might as well be described in Figure1.
Response 3: It has been reported elsewhere that extracellular matrix alters the drug sensitivity. The presence of extracellular matrix affects not only the polarity of the organoids, but also stimulates signalling pathways activated by cell-matrix interaction such as integrin mediated signalling. Uptake of drugs into cells can also be affected by extracellular matrix. Thus, multiple factors contribute to the drug sensitivity under the existence of extracellular matrix. It is intriguing to assess the contribution of polarity switch but we do not know it at this time. As mentioned in the response to Point2, whether CTOS organoids and CSC-organoids are interchangeable is also not yet clarified. This is mentioned in the legend of Figure 1.
Minor Points:
(1) Line 81 of Page2: "3.1.3. D culture" might be "3.1. 3D culture ".
(2) Line 69 of 5. Conclusion, "3.2. Figures and Tables" was erroneously inserted.
(3) All the references were numbered twice.
Response to minor points: All these points are corrected in the revised manuscript.
Reviewer 2 Report
See attachment

Author Response
We thank the reviewers for their constructive comments. We have carefully reviewed the comments, and respond as following. In this point-by-point response, to clearly distinguish our response from the reviewer’s comments, our responses were shown in blue texts. Also, in the main text, blue font was used to clearly indicate the revised portions of the manuscript.
Major Point 1: Paragraph 4.2 is a self-citation. Self-citation parts should be limited to the important aspect of their findings. They could also use a figure to describe their screening workflow and main hit results and I suggest to remand the readers to their original paper for further details. At the moment almost the entire 4.2 paragraph is dedicated only to their work. Are there other people using CTOS methods for drug screens?
Response 1: As far as we know, CTOS method has been used by other groups, but not been applied to drug screening. Therefore, the section 4.2 is a self-citation. Our screening workflow is described before citing our own works to facilitate the reader’s understanding. We did not add another figure to explain this because it is not that complicated.
Major Point 2: They mentioned that Organoid banks are available. Maybe could be useful for the readers if links are added on available banks, as they say are not many around the world
Response 2: We put the link to the largest organoid bank project directed by Hans Clevers, the inventor of CSC-organoid method.
Minor Points:
There is a typing mistake in the title of paragraph 3.1: it is written 3.1.3 D Culture … I guess should be 3.1 3D Culture ....
The typo is corrected in the revised manuscript.
A paragraph formatting problem is at the end of Conclusion section:’3.'2 Figure and Table" It is not clear to me where this line belongs?
This is inserted by mistake, and thus removed from the revised manuscript.
Reviewer 3 Report
The authors have done a good job on summarizing the different cancer models being used for drug screening and precision medicine. The aim of the paper highlights the importance/benefit of 3D model over 2D models in general.
The authors have touched on most of the 3D models established so far but defining 2D to conventional cell lines is bit misleading. And general statement claiming 3D being superior to 2D is not entirely true as translational value or predictive power of 3D models over 2D models is yet to be seen. On the other hand, it also depends on cancer types, for example 2D ex-vivo drug screening and its translational benefit on haematological cancer have already been established and being routinely used to explore therapeutics (repurposing) in treatment exhausted patients. Besides, there are several studies that highlights the translational potential/predictability of 2D patient derived cell lines screening (for eg.; conditionally reprogrammed cell lines, early passage PDX cell lines, primary cancer cells, co-culture with fibroblast and immune cells). They all need to be at least mentioned in brief and referenced.
On the 3D model part, it will be also worth mentioning/referencing ex-vivo explants culture, which poses immense potential as pre-clinical model.
It would be also nice to mention the limitations of organoids like lack of tissue microenvironment, heterogeneity between the organoids developed from different part of same tissue sample.
One minor comment, it is always good to use non-proprietary name of the compound if it exists rather than research code. Use “sepantronium bromide” instead of YM155.
Author Response
We thank the reviewers for their constructive comments. We have carefully reviewed the comments, and respond as following. In this point-by-point response, to clearly distinguish our response from the reviewer’s comments, our responses were shown in blue texts. Also, in the main text, blue font was used to clearly indicate the revised portions of the manuscript.
Major Points:
The authors have touched on most of the 3D models established so far but defining 2D to conventional cell lines is bit misleading.
To avoid the unwanted misunderstanding, we removed the statement of ‘established cancer cell lines in two-dimensional culture (hereafter referred to as “2D cell lines”)’, and changed it simply to ‘established cancer cell lines in two-dimensional culture’. The word, "established 2D cell line", was consistently used to indicate conventional cell lines cultured in 2D conditions and distinguished from cells originated from organoid culture or primary cells cultured in 2D conditions.
And general statement claiming 3D being superior to 2D is not entirely true as translational value or predictive power of 3D models over 2D models is yet to be seen.
It is true that the actual translational value of 3D over 2D is not yet entirely proven or disproven. Therefore, such explanation is added in the later part of section 4.1).
On the other hand, it also depends on cancer types, for example 2D ex-vivo drug screening and its translational benefit on haematological cancer have already been established and being routinely used to explore therapeutics (repurposing) in treatment exhausted patients.
We think that the culture of haematological cells is out of the argument of "2D vs 3D", because haematological cells are floating as single cells in vivo and thus physiological ex vivo or in vitro. In contrast, haematological cells adherent to plastic surface are, morphologically at least, non-physiological.
Besides, there are several studies that highlights the translational potential/predictability of 2D patient derived cell lines screening (for eg.; conditionally reprogrammed cell lines, early passage PDX cell lines, primary cancer cells, co-culture with fibroblast and immune cells). They all need to be at least mentioned in brief and referenced. On the 3D model part, it will be also worth mentioning/referencing ex-vivo explants culture, which poses immense potential as pre-clinical model.
As the reviewer suggested, it is worthwhile introducing these types of disease model. They are mentioned and referenced in the first paragraph of Chapter 3.
It would be also nice to mention the limitations of organoids like lack of tissue microenvironment, heterogeneity between the organoids developed from different part of same tissue sample.
The lack of tissue microenvironment is the limitation of ex vivo culture regardless of 2D or 3D. Thus, it is mentioned in the 3-2) Animal Models section that the presence of the microenvironment is the advantage of animal model over most of the ex vivo models.
The loss of intratumor heterogeneity is another issue common to any types of disease model using patient derived specimen. No matter how many samplings are performed for a tumor, it cannot cover the entire heterogeneity in the tumor. This can be a problem when applying the disease models for personalized medicine, because the presence of a resistant population or clone affect the actual drug response in patients even if it is a minor population in the tumor. On the other hand, it is ignorable when focusing on the drug development, and especially when building panels of cell, organoid, or PDX lines from multiple patients to study inter-tumor heterogeneity. This insight was pointed out in the section 4.4) Perspective for the Use of 3D Organoid Culture in Personalized Medicine.
Minor Point:
One minor comment, it is always good to use non-proprietary name of the compound if it exists rather than research code. Use “sepantronium bromide” instead of YM155.
As the reviewer suggested, we put the name "sepantronium bromide". We also left YM155 along with it, because YM155 is much more common than sepantronium bromide and it might be helpful for the readers.
Round 2
Reviewer 3 Report
I would like to thank and congratulate authors for the good work.